

# Removal or component reversal of local geomagnetic field affects foraging orientation preference in migratory insect brown planthopper *Nilaparvata lugens*

Yingchao Zhang[1,2] and Weidong Pan[1,2]

[1] Beijing Key Laboratory of Bioelectromagnetics, Institute of Electrical Engineering, Chinese Academy of Sciences, Beijing, China
[2] University of Chinese Academy of Sciences, Beijing, China

## ABSTRACT

**Background:** Migratory brown planthopper *Nilaparvata lugens* (*N. lugens*) annually migrates to Northeast Asia in spring and returns to Southeast Asia in autumn. However, mechanisms for orientation and navigation during their flight remain largely unknown. The geomagnetic field (GMF) is an important source of directional information for animals (including *N. lugens*), yet the magnetic compass involved has not been fully identified.

**Methods:** Here we assessed the influences of GMF on the foraging orientation preference of *N. lugens* by removing or component reversal of local GMF. At the same time, we examined the role of iron-sulfur cluster assembly1 (IscA1), a putative component of magnetoreceptor, in the foraging orientation preference of *N. lugens* under the controlled magnetic fields by RNA silencing (RNAi).

**Results:** We found that the near-zero magnetic field (NZMF) or vertical reversal of GMF could lead to *N. lugens* losing the foraging orientation preference, suggesting that a normal level of GMF, in the way of either intensity or inclination, was essential for the foraging orientation of *N. lugens*. Moreover, the gene knockdown of IscA1, also affected the foraging orientation preference of *N. lugens*, pointing out a potential role of IscA1 in the insects' sensing of variation in the GMF.

**Discussion:** These results suggested a foraging orientation preference is associated with the GMF and revealed new insights into the relationship between the IscA1 and magnetosensitivity mechanism in *N. lugens*.

Corresponding author
Weidong Pan, panwd@mail.iee.ac.cn

## INTRODUCTION

The brown planthoppers, *Nilaparvata lugens* (*N. lugens*), are recognized as a major migratory rice pest and virus vector. Adults exhibit wing dimorphism with macropterous and brachypterous phenotypes. The macropterous insects have functional wings for long-distance migration and the brachypterous individuals are non-migratory (*Guerra, 2011*). In East Asia, *N. lugens* adults overwinter in Vietnam and southern China. In order to find enough food and suitable living environment, they migrate to Northeast Asia in

spring and return back to Southeast Asia in autumn (*Kisimoto, 1976*; *Cheng et al., 1979*). *N. lugens* is a nocturnal species that usually takes flight sometime between sunset and sunrise (*Kisimoto, 1979*). However, the mechanisms for orientation and navigation during their flight remain largely unknown.

Many insects utilize magnetic information as a compass for their orientation and navigation. For instance, the migratory butterflies such as the sulphur butterflies *Aphrissa statira* or the monarch butterfly *Danaus plexippus* can orient with a sun compass, but are also observed migrating directionally under overcast skies. Accordingly, it has been confirmed that a magnetic compass was involved in both species (*Srygley et al., 2006*; *Guerra, Gegear & Reppert, 2014*). It is also the case for some migratory moths as the *Mythimna separata* that maintain migratory direction in the night sky (*Xu et al., 2017*). The use of magnetic compass has also been found in foraging of insects, including honeybee and ants. For honeybees, altering the GMF causes misdirection in the waggle dance, which is performed in the foraging trip, while there is no misdirection when dance orient along magnetic field lines (*Towne & Gould, 1985*). Ant foragers can be trained to recognize the location of a food source in the magnetic field and their orientation will turn according to the artificial magnetic fields (*Anderson & Meer, 1993*; *Camlitepe & Stradling, 1995*). In this respect, the use of magnetic compass as a key mechanism involved in insect directional movements appears feasible. This perspective was further supported by the findings of magnetic particles in insect tissues (*Gould, Kirschvink & Deffeyes, 1978*; *Wajnberg & Linhares, 1999*; *Chambarelli et al., 2008*; *Pan et al., 2016*), which could become the substrate for the magnetic compass. However, until now no behavioral observations have been confirmed the presence of magnetic compass in *N. lugens*.

So far there are two models which are most popular to explain how animals detect the magnetic field: the magnetite-based mechanisms (*Beason, 1986*; *Kirschvink & Gould, 1981*; *Lohman, 2010*) and the radical pair-based mechanisms (*Ritz, Adem & Schulten, 2000*). Recently, a light-magnetism-coupled magnetosensitivity model has been proposed, in which the homolog of the bacterial iron-sulfur cluster assembly, IscA1, that forms a complex with cryptochromes is suggested to serve as a putative magnetic protein biocompass (*Qin et al., 2016*). The iron-sulfur cluster proteins are ancient macromolecules with highly conserved structures. They have many functions including iron homeostasis, electron transfer, metabolic catalysis, nitrogen fixation, regulation of gene expression and the detection of reactive oxygen species (*Beinert, Holm & Münck, 1997*; *Beinert, 2000*). *Qin et al. (2016)* reported that the protein complex exhibited strong intrinsic magnetic polarity and rotated in synchrony with the external magnetic field. Previously we have found the IscA1 gene in *N. lugens* showed up-regulated mRNA expression during the period of migration (*Xu et al., 2017, 2019*). For the macropterous migratory *N. lugens*, compared with the GMF, the mRNA expression of the IscA1 gene and the cryptochrome1 gene were up-regulated under the magnetic fields of 0.5 millitesla (mT) and 1 mT in strength. The findings revealed that the expression of IscA1 and cryptochromes in *N. lugens* exhibited coordinated responses to the magnetic field, suggesting the potential associations between IscA1 and the magnetic sensory system.

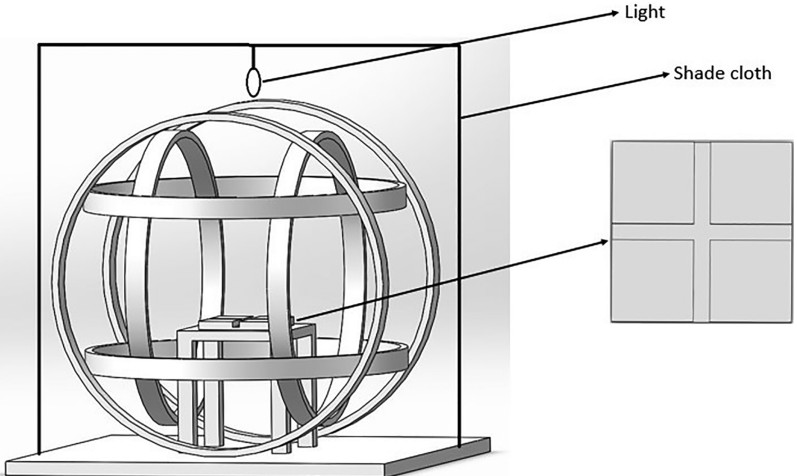

**Figure 1 The magnetic field generating device and the cross-tube choice chamber.** The Helmholtz coil system consisted of three independent coil pairs and each pair of coils was powered separately which could produce the magnetic field. The cross-tube choice chamber was placed horizontally inside the coil and a shade cloth covered the coil system during the experiment.

In this study, we demonstrated the effects of altered GMF, *i.e.*, near-zero magnetic field (NZMF) or components reversal of GMF, on the foraging orientation in *N. lugens*. By using the RNA silencing (RNAi) on *N. lugens*, the functional role of IscA1 was investigated.

# MATERIALS AND METHODS

## Insect stock

Experiments were performed at Beijing Key Laboratory of Bioelectromagnetics, Institute of Electrical Engineering, Chinese Academy of Sciences, Beijing, China. The insects were reared in climate chambers at day/night temperatures of $(27 \pm 1)$ °C/$(26 \pm 1)$ °C on susceptible Taichuang Native 1 (TN1) rice plant under 14:10 h light: dark cycle and $70 \pm 5\%$ humidity (*Wan et al., 2015*) and the environmental conditions of the chambers was the same in the entire experiment. The TN1 rice plants were prepared in advance, and used as the food for the insects when they grew up to 10 cm height. The migratory macropterous female and male adults were selected from the same generation for the successive generations (*Wan et al., 2016*).

## Magnetic field devices setup

The GMF used in the experiment (total intensity: 52,487 $\pm$ 841 nT; declination 5.30 $\pm$ 0.59°; inclination 56.29 $\pm$ 1.02°) were the local GMF at (39°59′14″N, 116°19′21″E). The artificial magnetic fields were produced using a Helmholtz coil system (Fig. 1). For NZMF, the Helmholtz coils were used to produce a near-zero magnetic field region with an average intensity of ~500 nT at a center spherical space (150 mm in radius). For component reversal of GMF, the Helmholtz coils were used to generate a magnetic field with twice intensity and reversed direction to offset either the horizontal component

or the vertical component of GMF, producing a reversed inclination with the same intensity but reversed component of GMF. Routinely before and after each experiment, we measured the three components of GMF using a fluxgate magnetometer (Model 191A, Honor Top Magnetoelectric Technology Co. Ltd., Qingdao, China; sensitivity: ±1 nT) to modulate the electric current of the coil pairs to produce the required intensity for NZMF.

## Cross-tube choice system and foraging orientation experiments

The choice system consisted of a cross tube which was embedded in a plastic square. The length of each arm of the cross tube was 110 mm. The width was 20 mm and the depth was 30 mm. The cross tube was covered with a plastic lid of same size. There were small holes at each arm end for air flowing through. A lamp (15 W, $\lambda$ = 320–680 nm) was installed as the light source (there is faint light when the *N. lugens* takes off at the sunset or sunrise) with 400 lumen of lumination intensity at the cross tube. The coil system was covered by a shade cloth during the experiment to shield the external environment (Fig. 1). During the experiment, the cross tube was placed horizontally inside the Helmholtz coils and the arms of the cross tube were oriented towards four cardinal points. The cardinal points used in the experiments were the same. Two cross tubes were used in the experiment, one containing food as a reward and the other without food. Ten fresh rice seedlings of susceptible variety of TN1 (*Fu et al., 2001*) was placed at one arm end as food reward.

The experiment was conducted in two parts: the first part of the trial was to provide food as a stimulus that the insects might associate with magnetic cues under the normal GMF, and the second part of the trial was to test for a disruption of their ability to exhibit this learned directional preference when the GMF was altered. Adult insects within 48 h of emergence regardless of gender or mating status were collected from the rearing colony and introduced into the center of the cross tube using a self-made insect suction-implement. The rice seedings were placed 70 mm away from the center (Fig. 2A). For the first part of the trial, the insects gathering around the rice seedlings (40 ± 13 insects) were eventually collected (Fig. 2B) using the same suction-implement into a vial and afterwards a new cross tube with no rice seedlings inside was placed horizontally in the Helmholtz coils. For the second part of the trial, the collected insects in the vial were replaced in the center of the new cross tube using the suction-implement (Fig. 2C) and the insects moving to each of the four arms of the new cross tube were recorded for 0.5 h (Fig. 2D). The entire experiment was performed at room temperature (26 ± 1 °C) and each magnetic field setup was performed individually for 12 replicates and the total number of insects tested was 511 ± 69.

## The effects of IscA1 gene silencing on the orientation of *N. lugens*

The IscA1 gene was previously cloned in *Nilaparvata lugens* and the results showed that the gene expression reached the peak at the third day after emergence (*Xu et al., 2017*), so adults of 1st day after emergence were selected for RNAi according to *Liu et al. (2010)*. The primers (Table 1) were designed based on the fragment sequence that was

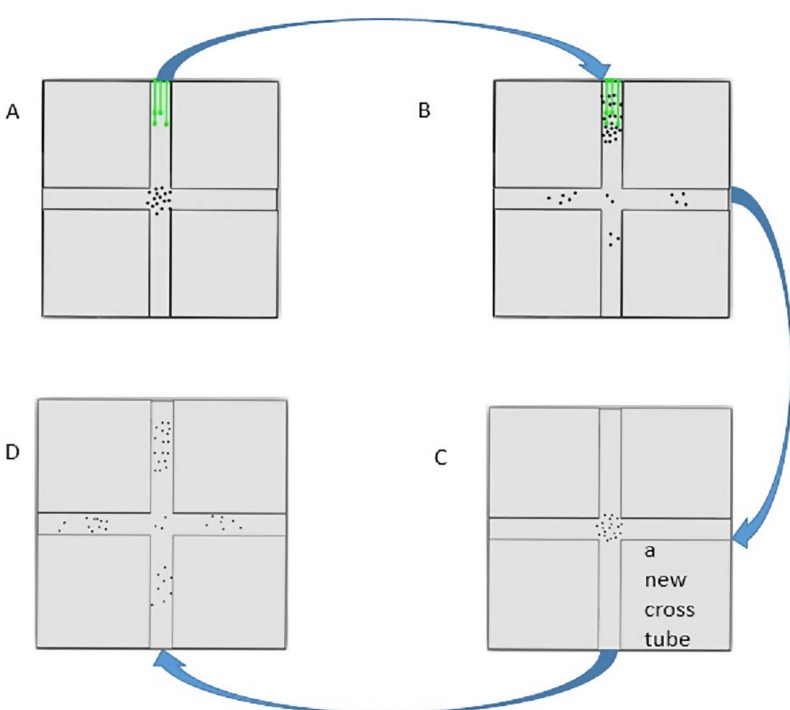

**Figure 2 The flow chart of cross-tube choice experiments.** (A) The insects were put in the cross tube with food source (green lines) inside. (B) Ten hours later, the insects gathered around in the food source were collected. (C) The collected insects were put in a new cross tube without food source. (D) After half an hour, the distribution of the insects was recorded.     

**Table 1 Primers used in the experiments.**

| Primer | Sequence (5′ to 3′) |
| --- | --- |
| NL-IscA1-dsRNA1-F | TAATACGACTCACTATAGGGAGAGCAGCACTGGTTTTGACAC |
| NL-IscA1-dsRNA1-R | TAATACGACTCACTATAGGGGCTTTTGCATCTATTATGACGA |
| NL-IscA1-dsRNA2-F | TAATACGACTCACTATAGGGGAGAAAGGAAAGTTTGACGAAG |
| NL-IscA1-dsRNA2-R | TAATACGACTCACTATAGGGAAAACTCCAAAGGAGAAAATGA |
| NL-GFP-dsRNA-F | TAATACGACTCACTATAGGGACGTAAACGGCCACAAGTTA |
| NI-GFP-dsRNA-R | TAATACGACTCACTATAGGGTGTTCTGCTGGTAGTGGTCG |
| NL-IscA-qPCR-F | AGACTCATATAGACGATGTTAA |
| NL-IscA-qPCR-R | CCTCCAATGTCCTCTAAC |
| NL-actin-qPCR-F | CCAACCGTGAGAAGATGACC |
| NL-actin-qPCR-R | GATGTCACGCACGATTTCAC |

searched from transcriptome of *N. lugens* by local BLAST search. The dsRNA of IscA1 gene was designed at two different regions: nearing the 3′end (dsRNA1) and nearing the 5′ end (dsRNA2). Insects were anesthetized with $CO_2$ for 30 s at $PCO_2 = 1$ mPa and immobilized on a 1.5% agarose plate with abdomen upward. Each insect was injected with 250 ng (50 nl in volume) dsRNA. On the second day after injection (24–48 h), the injected insects were collected and placed inside the GMF for foraging orientation test with the cross-tube system. The cross-tube behavioral trials for the RNAi insects were conducted in

the same manner as described in section 2.3 as part of a two-step trial. A total number of 700 insects were used for the experiments. To ensure the silencing efficiency, the expression level of *IscA1* gene was investigated before and after the behavioral test using three pools of six insects for each group by fluorescence-based quantitative real-time PCR (q-PCR) (*Bustin et al., 2009*). The whole body of adult *N. lugens* was used for sampling and all the samples were collected during the same time period (19:00–20:00 h). Total mRNA was extracted by TRIzol reagent (Invitrogen, Waltham, MA, USA). The quality of samples was determined by spectrophotometric optical density (OD) 260/280 and 2% agarose gel electrophoresis. The cDNA templates were synthesized with 1 μg of total RNA using PrimeScript™ RT reagent Kit with gDNA Eraser (TaKaRa, Tokyo, Japan). Each cDNA product was diluted with sterilized double- distilled water. The house-keeping gene for the q-PCR was *actin1* (GenBank accession No. EU179846), and the PCR amplification efficiency was established by means of calibration curves (*Bustin et al., 2009*). The optimized thermal program was designed according to the kit instructions. Quantification of the transcript level of genes was conducted according to the $^{\Delta\Delta}$Cq method (*Livak and Schmittgen, 2001*). RNA samples were analyzed independently for three times. The dsRNA of green fluorescent protein-GFP (GenBank accession No. U76561) was injected into the *N. lugens* as the control.

## Statistical analysis

All data were analyzed using SPSS 20.0 (IBM Inc., Armonk, NY, USA.). The *chi-square test* was used to analyze the ratio of the distribution of insects in four directions. If there was significant difference, Bonferroni correction was used to analyzed the difference between every two directions. One-way ANOVA was used to analyze the gene expression. Significant differences between *dsGFP* (control) and *dsNl-IscA1* injection treatments were determined by one-way ANOVA at $p < 0.05$.

## RESULTS

### The foraging orientation preference of *N. lugens* in the GMF *vs* NZMF

In the GMF, *N. lugens* showed the highest preference of foraging orienting to the north direction with original food ($\chi^2 = 108.48$, $p < 0.001$). The percentage of individuals orienting to the north was 39.06%, which was significantly higher than that to south 14% ($\chi^2 = 98.481$, $p < 0.001$), west 25.40% ($\chi^2 = 26.169$, $p < 0.001$) and east 21.39% ($\chi^2 = 45.652$, $p < 0.001$, Fig. 3A). In the NZMF, however, *N. lugens* were relatively equally distributed and the percentage of individuals orienting to the north, south, west and east direction was 25.46%, 22.22%, 27.55% and 24.77%, respectively ($\chi^2 = 10.261$, $p = 0.088$) (Fig. 3B).

### The foraging orientation preference of *N. lugens* in the horizontal or vertical component reversal of GMF

In order to study how the GMF affects the foraging orientation ability of *N. lugens*, we conducted behavior experiment in the horizontal or vertical component reversal of GMF.

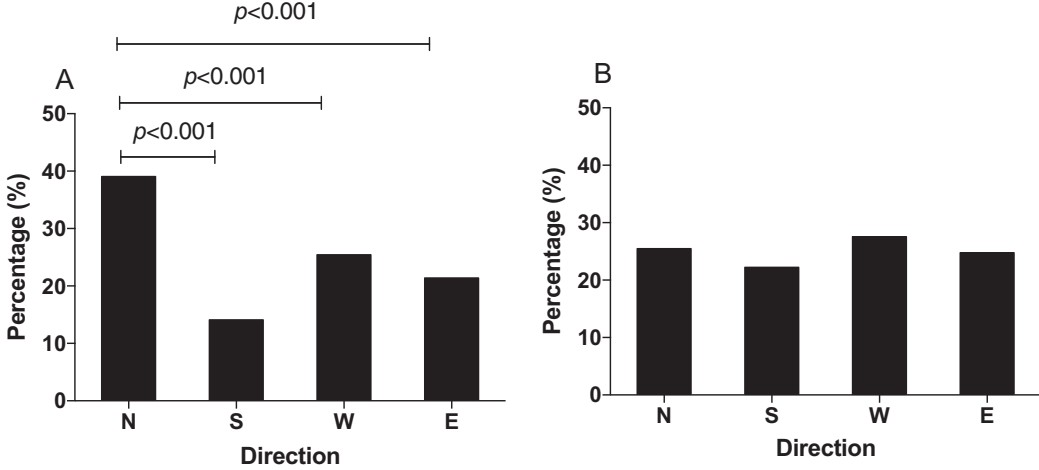

**Figure 3 The distribution of *N. lugens* in local geomagnetic filed (GMF, A) and near-zero magnetic field (NZMF, B) with food source initially located in the north direction.** A total number of insects with $N$ = 241, 87, 157, 132 for the distribution of insects in GMF and $N$ = 110, 96, 119, 107 for the distribution of insects in NZMF were used for experiments. Different lowercase letters indicate significant differences among directions by chi-square test at $p < 0.05$. Full-size  DOI: 10.7717/peerj.12351/fig-3

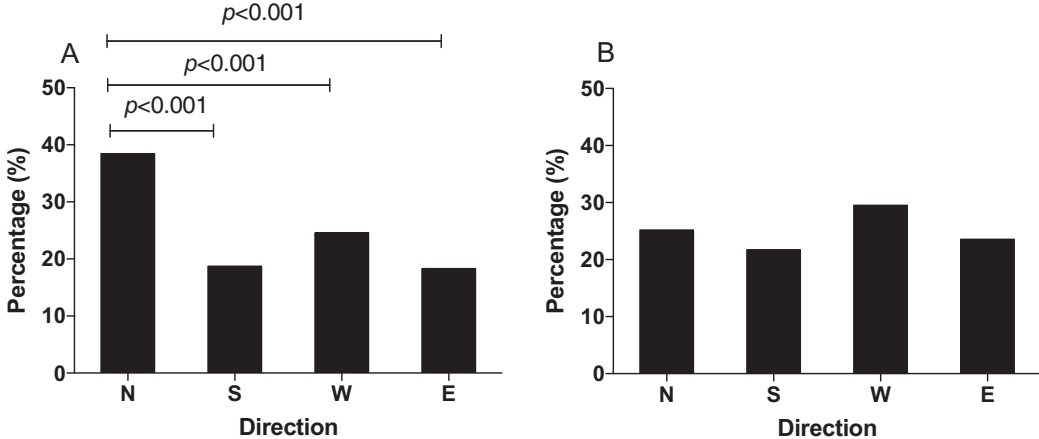

**Figure 4 The distribution of *N. lugens* under horizontal or vertical component reversal of local geomagnetic field (GMF) with food source initially located in the north direction.** (A) Horizontal component reversal of GMF. (B) Vertical component reversal of GMF. A total number of insects with $N$ = 191, 93, 122, 91 for the distribution in horizontal component reversal of GMF and $N$ = 139, 120, 163, 130 for the distribution in vertical component reversal of GMF were used for experiments. Different lowercase letters indicate significant differences among directions by chi-square test at $p < 0.05$.



In the horizontal component reversal of GMF, most of *N. lugens* were distributed in the north ($\chi^2$ = 87.872, $p < 0.001$, Fig. 4A), similar to that in GMF. In the vertical component reversal of GMF, the percentage of insects orienting to the north, south, west and east direction observed as 25.18%, 21.74%, 29.53% and 23.55%, respectively ($\chi^2$ = 9.371, $p$ = 0.102) (Fig. 4B).

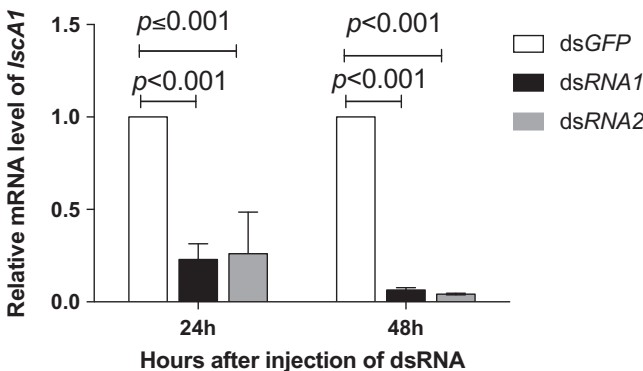

**Figure 5 The RNAi efficiency for IscA1 within 48 h after microinjection.** The relative expression level was quantified with regard to the *Nl-IscA1* value injected with *dsGFP*. The levels are expressed as means ±SE. Significant differences between *dsGFP* and *dsNl-IscA1* injection treatments were determined by one-way ANOVA at $p < 0.001$. Data points are means of three independent experiments.

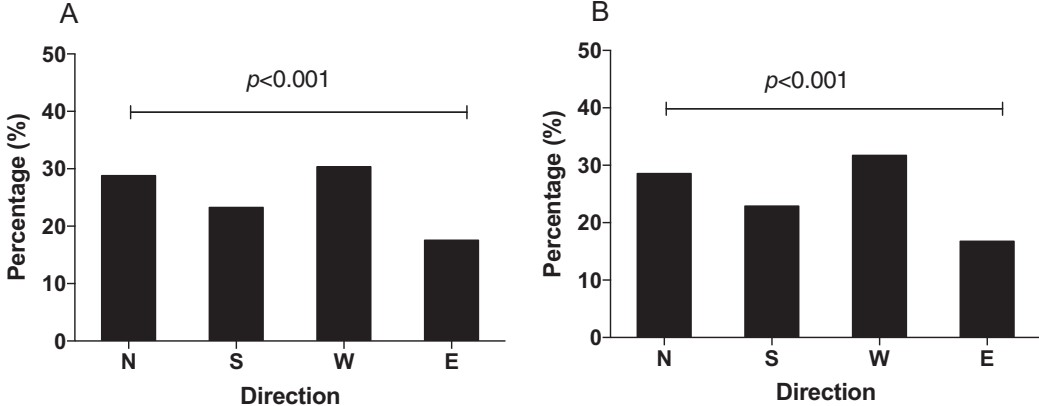

**Figure 6 The distribution of *N. lugens* with the IscA1 gene knockdown under local GMF (A) and horizontal component reversal of GMF (B) with food source initially located in the north direction.** A total number of insects with N = 151, 122, 159, 92 for the distribution of IscA1 gene knockdown in the north, south, west and east under GMF and insects with N = 126, 101, 140, 74 for the distribution of IscA1 gene knockdown in the north, south, west and east under horizontal component reversal of GMF were used for experiments.

## The IscA1 gene knockdown affected the foraging orientation preference of *N. lugens*

The q-PCR results showed that the mRNA expression of IscA1 was effectively downregulated after the gene silencing. Both of the silencing efficiencies of dsRNA1 and dsRNA2 were over 80% within 24 h (before the behavioral experiment) and 48 h (after the behavioral experiment) after microinjection (Fig. 5). Since the *N. lugens* preferred the north foraging direction in GMF or horizontal component reversal of GMF, we chose these conditions to investigate whether IscA1 gene silencing would affect the insects' choice of direction. In these two magnetic fields, most of the *N. lugens* with IscA1 gene silencing distributed in the west, followed by the north, south and east (Fig. 6). Compared with the wild type, the percentage of IscA1 gene knockdown *N. lugens* distributed in the

north direction revealed significantly decreased from 39.06% to 28.82% in GMF ($\chi^2$ = 13.183, $p$ < 0.001, Fig. 6A) and to 28.57% in the horizontal component reversal of GMF ($\chi^2$ = 10.151, $p$ < 0.001, Fig. 6B).

## DISCUSSION

Previous studies revealed that exposure of both small brown planthopper and brown planthopper to NZMF delayed egg and nymphal developmental durations and decreased adult weight and female fecundity of insects (*Wan et al., 2014*, *2021*). In addition to growth and development, the NZMF also affected positive phototaxis and flight capacity of the white-backed planthopper *Sogatella furcifera* (*Wan et al., 2016*, *2020*). Exposure to enhanced GMF also reduced the phototaxis of *N. lugens* (*Zhang et al., 2019*). In this study, we found majority of the insect *N. lugens* initially tested preferred north foraging orientation in the GMF, which was consistent with the field observation that *N. lugens* migrate to Northeast Asia under spring/summer-like conditions (*Kisimoto, 1976*; *Cheng et al., 1979*). In our experiment, the first part of the trial with rice seedlings to the north is to provide the opportunity for the insects to associate magnetic field information under the normal GMF with the presence of food in a particular direction, and the second part was designed to test whether the changed GMF (either NZMF in Fig. 3 or component reversal in Fig. 4) affected their ability to exhibit this learned directional foraging preference. As *S. furcifera* and *N. lugens* are both migratory insect pests of rice crops, the reported effects of removal of GMF suggest a role of the GMF, in terms of energy regulation or flight orientation, in their local scale foraging movement and also possibly their long-distance migration.

Generally, the inclination compass worked when the vertical component of the geomagnetic field was reversed, as it was shown that the mealworm beetle *Tenebrio molitor* significantly turned their preferred direction by 180° when the vertical component was reversed (*Vácha, tková Drš & žová Pů, 2008*). It has also been reported that birds could not distinguish between north and south by the polarity of the geomagnetic field, but could distinguish poleward and equatorward movement by the inclination of the field lines (*Wiltschko & Wiltschko, 1996*). In this study, when the vertical component of GMF was reversed, *N. lugens* individuals showed no significant foraging orientation. Thus, the foraging orientation of *N. lugens* in the vertical component reversed magnetic field was partially consistent to the inclination compass observed in monarch butterfly (*Guerra, Gegear & Reppert, 2014*). In this study, when the vertical component was reversed, *N. lugens* didn't distribute in the opposite direction as did the monarch butterfly. We speculate that this may be due to the existence of multiple compasses involved in insect orientation. *N. lugens* migrates in the sunset or sunrise (*Kisimoto, 1979*), so it's likely that a light-based mechanism of magnetoreception is also involved. Here our results suggested that a magnetic compass aided the foraging orientation preference of the migratory insect *N. lugens*, but *N. lugens* might also use other orientation cues.

As the homologue of bacterial iron-sulfur cluster assembly, the IscA1 has been found in most prokaryotic and eukaryotic organisms with highly conserved structures.

The inhibition of IscA1 could disrupt circadian rhythms in the fruit fly (*Mandilaras & Missirlis, 2012*). Moreover, it was found that knockdown of the IscA1 led to anemia in zebra fish (*Nilsson, Schultz & Pierce, 2009*). Currently a protein complex formed by the IscA1 interacting with cryptochromes was proposed as a putative magnetoreceptor and the protein crystal was claimed to exhibit strong magnetic polarity in response to an external magnetic field (*Qin et al., 2016*). The findings, however, have raised considerable controversy due to the broad interpretation of its biological meaning as well as the limitation of *in vitro* experiments (*Friis, Sjulstok & Solov'Yov, 2017*; *Hochstoeger, Nimpf & Keays, 2016*; *Pang et al., 2017*). Therefore, an independent investigation should be performed to clarify as far as possible whether the IscA1 is involved in specific processes of magnetosensitivity in terms of functional behaviors as navigation and orientation in long-distance migration of animals (*Meister, 2016*). In this study, our results showed that the foraging orientation preference of insects was affected by the IscA1 gene knockdown under varying GMF, providing direct evidence of IscA1 involved in magnetosensitivity of *N. lugens*. Meanwhile, biogenic magnetic particles were proposed to function as a hypothetic magnetoreceptor: the external magnetic field can affect the internal magnetite clusters leading to magnetic orientation loss (*Davila et al., 2005*). Previously we have detected magnetic particles in *N. lugens* (*Pan et al., 2016*) which also provides additional support for a magnetic sense in *N. lugens*. Whether the IscA1 protein is functionally linked to formation of magnetic particles and how these hypothetic magnetoreceptors work in synergism *in vivo* remains to be further elucidated.

## CONCLUSION

This study provided behavioral evidence that the foraging orientation preference of the migratory insect *N. lugens* is affected by removal or component reversal of local GMF. When the vertical component of GMF was reversed, the insects showed no significant foraging orientation preference, suggesting the potential use of inclination compass-aided orientation in *N. lugens*. The foraging orientation preference of *N. lugens* was also affected by the IscA1 gene knockdown, providing a feasible mechanistic explanation for the insects' sensing of variation in the GMF. Further work is needed to investigate the potential associations between the IscA1 and magnetic particles in terms of the magnetosensitivity mechanism in *N. lugens*.

## ACKNOWLEDGEMENTS

We thank Prof. Hongxia Hua of College of Plant Science and Technology of Huazhong Agricultural University providing the original laboratory stock of the *N. lugens*. We also thank Space Electromagnetic Environment Laboratory for providing geomagnetic data.

### Funding

This research was supported by the National Natural Science Foundation of China (31670855, 31870367, 31470454, 31672019, 31701787), the National Department of Public

Benefit Research Foundation (201403031), the Natural Science Foundation of Jiangsu Province Youth Fund (SBK2016043525) and the China Postdoctoral Science Foundation (2016M590470). The funders had no role in study design, data collection and analysis, decision to publish, or preparation of the manuscript.

## Grant Disclosures

The following grant information was disclosed by the authors:
National Natural Science Foundation of China: 31670855, 31870367, 31470454, 31672019, 31701787.
National Department of Public Benefit Research Foundation: 201403031.
Natural Science Foundation of Jiangsu Province Youth Fund: SBK2016043525.
China Postdoctoral Science Foundation: 2016M590470.

## Competing Interests

The authors declare that they have no competing interests.

## Author Contributions

- Yingchao Zhang conceived and designed the experiments, performed the experiments, analyzed the data, prepared figures and/or tables, authored or reviewed drafts of the paper, and approved the final draft.
- Weidong Pan conceived and designed the experiments, analyzed the data, authored or reviewed drafts of the paper, and approved the final draft.

## DNA Deposition

The following information was supplied regarding the deposition of DNA sequences:
The green fluorescent protein-GFP sequence is available at GenBank: U76561.

## Data Availability

The raw data are available in a Supplemental File.

## Supplemental Information

Supplemental information for this article can be found online at http://dx.doi.org/10.7717/peerj.12351#supplemental-information.

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
