# Peer review of "Removal or component reversal of local geomagnetic field affects foraging orientation preference in migratory insect brown planthopper Nilaparvata lugens"

_PeerJ, doi:10.7717/peerj.12351_

## Round 0.1 · original submission · Major Revisions

Dear Dr. Zhang and Pan:

Thanks for submitting your manuscript to PeerJ. I have now received three independent reviews of your work, and as you will see, the reviewers raised some concerns about the research. Despite this, these reviewers are optimistic about your work and the potential impact it will have on research studying foraging behavior in the brown planthopper. Thus, I encourage you to revise your manuscript, accordingly, taking into account all of the concerns raised by both reviewers.

While the concerns of the reviewers are relatively minor, this is a major revision to ensure that the original reviewers have a chance to evaluate your responses to their concerns. There are many suggestions, which I am sure will greatly improve your manuscript once addressed.

Importantly, please ensure that an English expert has edited your revised manuscript for content and clarity.

Please revise your experimental design for clarity. Your methods should be clearly outlined, and your experiments should be repeatable.

Please also not that reviewer 1 has included a marked-up version of your manuscript.

Therefore, I am recommending that you revise your manuscript, accordingly, taking into account all of the issues raised by the reviewers.

I look forward to seeing your revision, and thanks again for submitting your work to PeerJ.

Good luck with your revision,

-joe

Reviewer 1 ·

Basic reporting

no comment

Experimental design

no comment

Validity of the findings

no comment

Additional comments

The manuscript titled "Removal or component reversal of local geomagnetic field affects foraging orientation preference in migratory insect brown planthopper Nilaparvata lugens" is a very interesting study about the effect of IscA1 gene on magnetoreception and deserves to be published in PeerJ. However, there are some errors and shortcomings, which are following as:
Q1: Abstract: Methods: L23: “in magnetic field” should be changed as “under the controlled magnetic fields”.
Q2: Abstract: Results: L23: L27-28: “...also affected the foraging orientation preference of N.
Lugens” What about the magnetic fields?
Q3: Fig.2: In the center of the cross-tube, no individuals of N. lugens remained (i.e., nonmoving individuals”? e.g., L148-150: The percentage of insects orienting to the north was 39.06%, which was significantly higher than that in south 14%, west 25.40% and east 21.39% (Fig. 3A). Total 99.85% individuals moved, and 0.15% individuals non-moved!
Q4: Fig.3, Fig.4 and Fig.6: Why did you supply food source located in the north direction of the cross-tube, not S, W or E?
Q5: There were some errors of languages and writing etc., e.g., in L16, "return" (returns);in L21, "reversing" (reversal); in L37, delete “which”. Other corrections were revised and marked directly in the PDF file.
Q6: The result about the foraging orientation under reversal of the vertical component is quite interesting and distinguishable from previous study of planthopper N. lugens. Also please supplement the references by the newly published paper about altered migratory traits under changed GMF in the insect Ni. lugens by Wan et al. (2020a, 2020b).
Wan GJ et al. 2020a. Geomagnetic field absence reduces adult body weight of a migratory insect by disrupting feeding behavior and appetite regulation. Insect Science (https://doi.org/10.1111/1744-7917.12765)
Wan GJ, et al. 2020b. Geomagnetic field intensity as a cue for the regulation of insect migration. Biological Letter (http://dx.doi.org/10.1098/rsbl.2019.0940)

Annotated reviews are not available for download in order to protect the identity of reviewers who chose to remain anonymous.

Reviewer 2 ·

Basic reporting

First, overall the writing and figure presentation is quite well. In the following, I will propose some suggestions to improve the manuscript. The formatting of the manuscript is not uniform. I would suggest using uniform justified as style.
1. In page 3, add some transitional phrases between 1st and 2nd paragraph for a better smooth context.
2. In page 3, line 54: use full scientific names for the first time, and use abbreviations for the rest of generic names.
3. In page 3, line 54-55: list literature in the order of publication, and sic passim.
4. In page 3, line 58: change the word to either singular or plural form.
5. In page3, line 64-65: the phrase “since other mechanisms of navigation, foraging or homing are unavailable” is a bit arbitrary and should be modified appropriately.
6. In page4, line 83: check the publication date of literature.
7. In page5, line 98: change “27±1°C/26±1°C” to “(27±1) °C/(26±1)°C”.
8. In page5, line 102: should it be described as length or height for the rice seedings?
9. In page5, line 107-108: check whether the latitude and longitude symbols are correct.
10. In page6, line 123: indicate lumination intensity at the cross tube.
11. In page6, line 132: indicate detailed information of test insects such as age, sex or mating status.
12. In page6, line 140: indicate environmental conditions at experiments.
13.In page6, line 143: check the literature.
14. In page7, line 160: describe how the One-way ANOVA was used.
15. In page7, line 166, 170: mark the Chi-square value and P-value even it is not significant.
16. In page8, line 181: change the phrase “The foraging orientation preference of insects as influenced by the IscA1 gene knockdown” to “The IscA1 gene knockdown affected the foraging orientation preference of insects”.
17.In page9, line 197: add the literature.
18. In page9, line 203-209: the phrase should be reorganized.
19. In page11, line 251: the conclusion should better be reorganized for concision.
20. In page12, line 290: check all the references for a uniform style, and a correct citation for the literature “Cheng HN, Chen JC, His H, Yang LM, Chu TL, Wu CT, Chen JK, Yang CS. 1979. Studies on the migrations of brown planthopper Nilaparvata lugens Stal. Dialogue, 33, 164-167. DOI:10.1017/S012217300038932.”

Experimental design

1. In the behavioral test, the physiological state of the tested insect should be described in detail, such as gender, age, mating experience, etc.
2. The q-PCR experiments has not been carried out following the highest standards (see the MIQE rules). And also add the reference gene for the q-PCR experiment and the calculation formula.

Validity of the findings

This manuscript demonstrated that the near-zero magnetic field (NZMF)or vertical reversal of geomagnetic field (GMF) could lead to brown planthopper Nilaparvata lugens losing the foraging orientation preference, suggesting that a normal level of GMF was essential for the foraging orientation of N. lugens. Also, the authors silenced N. lugens IscA1 resulted in affected orientation preference of N. lugens, pointing out a potential role of IscA1 in the insects’ sensing the varying GMF. This observation suggests that a foraging orientation preference is associated with the GMF, and reveals new insights into the relationship between the IscA1 and magnetosensitivity mechanism in N. lugens. This result is very interesting, and the study seems to be well executed.

Additional comments

The authors should comment a little more about the experimental setup in the discussion as to connect the results with a possible interpretation involved in the individual test design.

·

Basic reporting

The English was understandable, but could use some additional final proofreading to avoid confusion and improve clarity. I did not do a detailed proofreading of the text, but rather tried to focus on the scientific content.

Structure of the article and the flow of the information presented was appropriate.

Table 1 providing the q-PCR primer information was not cited in the text of the article.

Figures 3, 4 and 6 – The use of patterns/shading in the bar graphs is unnecessary. The bars are distinguished by the labels on the X-axis. The shading does not provide any additional information needed to interpret the graph.

Line 53. I think something is missing here. Did you instead intend to say altering the GMF causes misdirection?

Experimental design

The research question was well-defined. The experimental outcomes contribute to the understanding of responses of insects to magnetic stimuli and their use as environmental ques for movement, along with the underlying mechanisms involved in magnetoreception.

The protocols used to impose modifications of the geomagnetic field as treatments are well-developed and have been used to study insect physiological and behavioral responses to magnetic stimuli in several prior studies.

In several cases, more methodological detail is needed to better describe how the experiments were conducted and properly interpret the results.

Line 118. Please provide more detail about how the insects were introduced into the center of the cross-tube arena. Where they placed in individually? On an infested plant? Or perhaps collected from the rearing colony into a vial or cup that was then place in the center of the arena?

Lines 116-118. It is not clear where the food (rice seedlings) was placed during the initial choice trials. After reading the Results, it seems as though the plants were placed ONLY at the end of the north tube? If so, then I assumed that the purpose of this first part of the trial was to provide food as a stimulus that the insects might associate with magnetic cues under the normal GMF, and the purpose of the second part of the trial was to test for a disruption of their ability to exhibit this learned directional preference when the GMF was altered. In addition to clarifying the position of the rice seedlings during the first trial, I suggest that you also briefly explain why there were two steps involved in the assays.

Line 134. Were the cross-tube behavioral trials for the RNAi insects conducted in the same manner as described in section 2.3 as part of a two-step trial? Please clarify or explain how the assay was modified for the RNAi insects.

Lines 135-139. The description of the qPCR methods is incomplete. What were the housekeeping gene(s)? Has the expression level of the housekeeping genes been tested to be sure the expression level is constant when exposed to different GMF treatments? How was RNA extracted? Table 1 containing the primer information was not cited in the text.

Validity of the findings

The interpretation of the data seems to be confusing the use of magnetic stimuli as a cue involved in local foraging movements versus long distance migratory flights. The relevance of the study results to local foraging movements versus long distance migratory movements needs to be clarified in the text on lines 184-185 and 190-191.

Strictly speaking, the data shown in Figures 3 and 4 are most relevant to local scale movements associated with foraging for food. Related to my comment above about the behavioral assay methods (lines 116-118), I interpreted the intent of the first part of the assay with food to the north to provide the opportunity for the insects to associate magnetic field information under the normal GMF with the presence of food in a particular direction, and the second part was intended to test whether changing the magnetic field either by reducing it (NZMF, Figure 3) or altering it (component reversal, Figure 4) affected their ability to exhibit this learned directional foraging preference. Provided I did not misunderstand how and why the experiment was conducted, then interpretation of these results is directly related to the use of GMF stimuli in local scale foraging, not longer distance migratory flight. It doesn’t rule out the use of GMF in long distance flight, and indicates that the insects are able to detect and respond to GMF changes as imposed by your treatments, but your data better address GMF as a cue involved in local movement - which is nonetheless still very interesting.

Additional comments

All my comments are included in the other sections,

---

## Round 0.2 · Minor Revisions

Dear Dr. Zhang and Pan:

Thanks for revising your manuscript. The reviewers are mostly satisfied with your revision (as am I). Great! However, there are a few concerns to address. Please attend to these issues ASAP so we may move towards acceptance of your work.

Best,

-joe

Reviewer 1 ·

Basic reporting

As it is shown, the manuscript titled "Removal or component reversal of local geomagnetic field affects foraging orientation preference in migratory insect brown planthopper Nilaparvata lugens" has been addressed substantially for content and clarity by the authors, and this version has been revised based on the comments of the reviewers. I am pleased to recommend its publication in PeerJ.

Experimental design

no comment

Validity of the findings

no comment

Additional comments

no comment

·

Basic reporting

The authors addressed the points raised in my previous review.

Experimental design

The authors addressed the points raised in my previous review.

Validity of the findings

The authors addressed the points raised in my previous review.

Additional comments

I have attached an edited version of the manuscript with suggested grammatical corrections that I hope will be helpful in preparing the final version.

---

## Round 0.3 · accepted · Accept

Dear Dr. Zhang and Pan:

Thanks for revising your manuscript based on the concerns raised by the reviewers. I now believe that your manuscript is suitable for publication. Congratulations! I look forward to seeing this work in print, and I anticipate it being an important resource for groups studying foraging behavior in the brown planthopper. Thanks again for choosing PeerJ to publish such important work.

Best,

-joe